# Cancer Incidence in Korean Healthcare Workers in Hospitals

**DOI:** 10.3390/cancers15072045

**Published:** 2023-03-29

**Authors:** Dong-Wook Lee, Hyeonjun Kim, Wanhyung Lee, Woo-Ri Lee, Ki-Bong Yoo, Jun-Hyeok Choi, Kyung-Eun Lee, Jin-Ha Yoon

**Affiliations:** 1Department of Occupational and Environmental Medicine, Inha University Hospital, Inha University, Incheon 22332, Republic of Korea; 2Jungbu Korea Occupational Diseases Surveillance Center, Incheon 22332, Republic of Korea; 3Department of Occupational and Environmental Medicine, Chonnam National University Medical School and Chonnam National University Hwasun Hospital, Hwasun-gun 58128, Republic of Korea; 4Department of Occupational and Environmental Medicine, Gil Medical Center, Gachon University College of Medicine, Incheon 21654, Republic of Korea; 5Department of Research and Analysis, National Health Insurance Service Ilsan Hospital, Goyang-si 10444, Republic of Korea; 6Division of Health Administration, College of Software and Digital Healthcare Convergence, Yonsei University, Wonju 26426, Republic of Korea; 7Occupational Safety and Health Research Institute, Korea Occupational Safety and Health Agency, Ulsan 44429, Republic of Korea; 8Department of Occupational and Environmental Medicine, Severance Hospital, Yonsei University College of Medicine, Seoul 03722, Republic of Korea

**Keywords:** healthcare workers, cancer incidence, breast cancer, hospital

## Abstract

**Simple Summary:**

Investigation of the excess cancer risk among healthcare workers in hospitals is crucial. We constructed a large, nationwide retrospective cohort including 107,646 healthcare workers in hospitals. We found significantly increased standardized incidence ratio (SIR) of all-cancer (SIR = 1.25, 95% confidence interval [CI] = 1.06–1.47) and breast cancer (SIR = 1.21; 95% CI = 1.09–1.36) among female healthcare workers. These results imply that potential carcinogens of hospital workers should be assessed, especially female workers in the hospital, including nurses.

**Abstract:**

Objectives: Healthcare workers in hospitals (HHCWs), a notably increasing workforce, face various occupational hazards. A high incidence of cancer among HHCWs has been observed; however, the cancer incidence status among HHCWs in South Korea is yet to be studied. This study aimed to assess cancer incidence among HHCWs in South Korea. Methods: We constructed a retrospective cohort of HHCWs using National Health Insurance claims data, including cancer incidence status and vital status, from 2007 to 2015. Those who had worked in hospitals for at least three years were defined as HHCWs. Standardized incidence ratios (SIRs) for all cancer types and standardized mortality ratios were calculated. Results: A total of 107,646 HHCWs were followed up, and the total follow-up duration was 905,503 person-years. Compared to the total workers, female HHCWs showed significantly higher SIR for all cancers (observed cases = 1480; SIR = 1.25; 95% confidence interval [CI] = 1.06–1.47). The incidence of breast cancer among female HHCWs was significantly higher compared to that among total workers (observed cases = 376; SIR = 1.21; 95% CI = 1.09–1.36). Conclusions: Our findings indicate that female HHCWs have an elevated probability of developing cancer, which suggests that occupational risk factors such as night-shift work, anti-neoplastic medications, stressful jobs, and ionizing radiation should be assessed. Further investigation and occupational environment improvement activities are required.

## 1. Introduction

Healthcare workers (HCWs) constitute a fast-growing sector of the workforce in most countries, and this trend has accelerated since the coronavirus disease 2019 pandemic. Health professionals, health associate professionals, personal care workers in health services, health management and support personnel, and other health service providers can be classified as HCWs who provide care and services to patients [1]. According to the Ministry of Employment and Labor of Korea, the total number of business employees in 2019 was 18,743,650. Among them, the total number of workers who worked in industries classified as hospitals was 571,482, accounting for approximately 3% of the total workers [2].

The workplace of HCWs can be categorized as clinics and hospitals. A clinic provides outpatient medical services and is smaller than a hospital. A hospital provides inpatient care with or without emergency care and specialist care, or surgery, for more serious life-threatening conditions. According to the medical law of South Korea, a hospital is defined as an institution with at least 30 beds, where doctors (and/or dentists) mainly provide medical care for inpatients [3].

In South Korea, most workers in hospitals are HCWs, including doctors, nurses, nursing assistants, radiologists, medical technicians, and various other workers, such as cleaners and cooks. Most HCWs in South Korea work in hospitals rather than clinics. According to the data reported by the Korean National Health Insurance Service in 2022, the number of healthcare workers in hospitals (HHCWs) was 562,117, and the number of HCWs in clinics was 219,689 [4]. HHCWs mainly include registered nurses (222,746, 39.6%), assisting nurses (78,349, 13.9%), doctors (61,706, 11.0%), physical therapists (20,876, 3.7%), clinical pathologists (19,116, 3.4%), radiologists (18,932, 3.4%), and other workers in offices (92.977, 16.5%).

HCWs can be exposed to a variety of physical, chemical, and psychosocial occupational hazards; in particular, some exposures are also known to be potential carcinogens to humans, including psychosocial work environments (long working hours and shiftwork), biological hazards such as viruses, physical hazards such as diagnostic or therapeutic radiation, and chemical agents such as anti-neoplastic agents or formaldehyde. Previous studies reported high cancer incidence among HCWs [5,6,7]; however, the results were inconsistent. Furthermore, a standardized cancer incidence among HHCWs in South Korea has not been reported, although cancer in HCWs has been consistently focused on and issued in the occupational safety and health area [8,9,10]. This study aimed to investigate the high incidence of cancer among HHCWs compared with total workers and public officials.

## 2. Methods

### 2.1. Cohort Definition

We constructed a cohort using the National Health Insurance (NHI) claims data in South Korea, which was provided via the National Health Insurance System (NHIS). The NHS is a single-payer health insurance system in South Korea that is a nationally operated social security system. All residents of South Korea must have health insurance by law, and almost all people residing in the country are covered by NHI. All medical services covered by insurance are claimed by the NHI. Therefore, the NHIS contains information on the medical care of the entire population. The NHIS has data on the socioeconomic status of insurance beneficiaries to calculate and collect insurance premiums, including whether they are wage workers. The cohort comprised data on wage workers aged 25–65 years from 2007 to 2015, and this cohort included information on the industry classification of each worker. There were no missing values in the industry classification of the study participants. The industry classifications were based on the Korean Standard Industrial Classification [11].

### 2.2. Healthcare Workers in Hospital and Control Groups

The occupation of the study participants can be defined depending on the time point of observation and cohort design. The NHIS provides data on the industry to which the participant belongs, on an annual basis. In South Korea, a medical service facility that has ≥30 beds for patient admission is defined as a “hospital,” and that with <30 beds is defined as a “clinic.” In this study, we defined HHCWs as participants whose workplace was registered as a “hospital” for at least three consecutive years from 2007 to 2015 (“Fixed”). We defined two control groups to investigate whether cancer incidence and mortality increased among HHCWs compared to other workers. HHCWs were also included in the all-workers group to investigate whether HHCWs have an excess risk of cancer compared to the total workers in South Korea. Public officials in South Korea work in an environment relatively safe from occupational carcinogens; therefore, we set public officials as a healthy control group to compare excess cancer risk related to the occupation [12]. All workers and public officials who had three consecutive years of identical industrial classification from 2007 to 2015 were selected as the control group.

### 2.3. Cancer and Vital Status Identification

During the study period (2007–2015), patients with cancer were identified as those who visited the hospital with a diagnostic code of C00–C99 for cancer, according to the International Classification of Diseases, 10th revision. To investigate the cancer incidence, we set up a one-year washout period. That is, participants with a diagnostic code for cancer in the first year of follow-up were excluded. Hospital visits included any contact with the hospital through the emergency room, outpatient visits, inpatient admissions, or referrals from primary or tertiary healthcare providers. Cancer sites were classified as follows: [C00–C97] malignant neoplasm; [C00–C14] lip, oral cavity, and pharynx; [C15–C26] digestive organs; (C16) stomach; (C18) colon; (C22) liver and intrahepatic bile ducts; (C25) pancreas; [C30–C39] respiratory and intrathoracic organs; (C33–C34) trachea and malignant neoplasm of bronchus and lung; [C40–C41] bone and articular cartilage; [C43–C44] skin; [C45–C49] mesothelial and soft tissue; [C50–C50] breast; [C51–C58] female genital organs; [C60–C63] male genital organs; [C64–C68] urinary tract; [C70–C72] brain and other parts of central nervous system; [C81–C96] lymphoid; hematopoietic and related tissue (C82–C85); non-Hodgkin’s lymphoma; and (C91–C95) leukemia. Mortality cases were defined as participants who died before the end of follow-up (31 December 2015), identified using the mortality registration data of Statistics Korea.

### 2.4. Other Variables

Demographic variables of participants were collected from the NHIS database, including age (years), sex (male or female), and household income. To describe the distribution of the ages, we grouped participants as 25–34, 35–44, 45–54, and 55–64 years. The NHIS database provides household income in a range from 1 (lowest) to 20 (highest). Household income levels were divided into four groups (Q1: 1–5; Q2: 6–10; Q3: 11–15; Q4: 16–20).

### 2.5. Statistical Analysis

An indirect standardization method was used to calculate the standardized incidence ratios (SIR) using age-specific expected cases of the control group. We calculated the SIRs and standardized mortality ratios (SMRs) of HHCWs compared to all workers and public officials in South Korea. The observed cases of death were divided by the expected cases of death, with the standardization of sex and five-year age groups from 25 to 64 years. The age- and sex-specific mortality rates of all workers and public officials during the study period were used as a reference. The SIRs and SMRs were calculated in both cohorts for 21 specific outcomes. The Bonferroni method was used to control the overall testing error rate for multiple comparisons. A value of *p* < 0.002 was considered significant and was approximately calculated as 0.05 divided by 21. For the sensitivity analysis, we defined HHCWs differently from the main analysis; participants whose workplaces were registered as a “hospital” for at least one year were designated as HHCWs (“Dynamic”). All analyses were performed using SAS version 9.4 (SAS Institute, Cary, NC, USA). This study was approved by the Institutional Review Board at the Yonsei University (Y-2017-0100).

## 3. Results

Table 1 presents the descriptive statistics and characteristics of the study participants. A total of 107,646 HHCWs were included in this cohort. The total followed-up length was 905,503 person-years, and the mean followed-up person-years (±standard deviation (SD)) was 8.41 (±0.84) years. The total number of workers and public officials included in this study as the control groups to estimate SIRs were 8,317,710 and 611,833, respectively. Most of the HHCWs were females (65.4%) as opposed to males (34.6%). The percentages of males and females among the total workers and public officials were 69.0% and 31.0% and 49.0% and 51.0%, respectively. HHCWs were mostly in the age group of 25–34 years, as the proportions of HHCWs in the 25–34, 35–44, 45–54, and 55–64 years age groups were 48.7%, 28.6%, 16.1%, and 6.6%, respectively. The proportions of total workers and public officials in the 25–34, 35–44, 45–54, and 55–64 years age groups were 34.5%, 34.0%, 24.0%, and 7.6%, respectively, and 27.4%, 34.0%, 30.7%, and 7.8%, respectively. The income level of HHCWs was mostly in groups Q2 (35.0%) and Q3 (25.9%) and that of the total workers was relatively evenly distributed. Public officials showed higher income status, with 31.9% and 57.8% of them in groups Q3 and Q4, respectively.

Table 2 shows the SIRs of malignant neoplasms and all-cause SMRs of HHCWs compared to those of the total workers. SIRs of malignant neoplasms (C00–C97) were 0.83 (95% confidence interval [CI] = 0.73–0.96) and 1.25 (95% CI = 1.06–1.74) among men and women, respectively. The observed cases (Obs) and expected cases (Exp) of malignant neoplasm among women were 1480 and 1360.9, respectively, and the observed SIR of 1.25 (95% CI = 1.06–1.47, *p* = 0.001) was significantly higher than in the total workers. Female HHCWs showed higher SIRs of breast cancer (C50). The expected and observed cases of breast cancer among female HHCWs were 310 and 376, respectively, which resulted in an SIR of 1.21 (95% CI = 1.09–1.36, *p* < 0.001). The all-cause SMR was significantly higher among female HHCWs than that of all the workers (SMR = 1.25, 95% CI = 1.06–1.47, *p* = 0.0019).

In Table 3, the SIRs of malignant neoplasms and all-cause SMRs of HCWs compared to those of public officials are presented. SIRs of malignant neoplasms (C00–C97) were significantly higher at 1.15 (95% CI = 1.05–1.28, *p =* 0.001) and 1.15 (95% CI = 1.09–1.22, *p* < 0.001) among men and women, respectively. Among men, significantly higher SIRs than public officials were observed for respiratory and intrathoracic organs (SIR = 1.84, 95% CI = 1.29–2.79, *p* < 0.001), and non-Hodgkin’s lymphoma (SIR = 2.19, 95% CI = 1.14–4.89, *p* = 0.0017). Among women, the highest cancer incidence was observed for breast cancer (SIR = 1.18, 95% CI = 1.06–1.32, *p* = 0.001); followed by female genital organs (SIR = 1.49, 95% CI = 1.21–1.85, *p* < 0.001); mesothelial and soft tissue (SIR = 2.19, 95% CI = 1.16–4.75, *p* = 0.001); and digestive organs (SIR = 1.34, 95% CI = 1.15–1.58, *p* < 0.001).

Appendix A shows the characteristics of the cohort in the sensitivity analysis, which defined HHCWs as participants whose workplace was registered as a “hospital” for at least one year. In this dynamic cohort, the number of HCWs in the study period was 363,847, which was more than 3.38 times the number in the original cohort. The difference in number between the original and dynamic cohorts was also observed in the total workers and public officials, with a 1.32 and 1.10 times difference, respectively. In the sensitivity analysis (Appendix A), SIRs for breast cancer (SIR = 1.13, 95% CI = 1.03–1.25, *p* = 0.003) and mesothelial and soft tissue cancer (SIR = 1.84, 95% CI = 1.12–3.27, *p* = 0.002) in female HHCWs were marginally significantly higher than in the total workers. Cancers in male genital organs were significantly lower in male HHCWs than in public officials (SIR = 0.57, 95% CI = 0.42–0.79, *p* = 0.0016). All-cause mortality was significantly higher in male and female HCWs than in public officials, at SIR of 1.33 (95% CI = 1.16–1.55, 95% CI < 0.001) and 1.32 (95% CI = 1.16–1.52, *p* < 0.001), respectively.

Appendix A summarizes the significant findings of this study. All-cause death, malignant neoplasms, and digestive organ involvement were significantly higher in males and females. Excess SIRs in the respiratory and intrathoracic organs, trachea, malignant neoplasms of the bronchus and lung, and non-Hodgkin’s lymphoma were observed in males. In females, excess SIRs were observed in the breast, mesothelial and soft tissue, and female genital organs.

## 4. Discussion

We constructed a retrospective cohort of 107,646 HHCWs, with a total follow-up duration of 905,503 person-years. Compared to the total workers, female HHCWs showed a significantly higher SIR for all cancers. Breast cancer was the most common cancer and was more frequently observed among female HHCWs than among all workers and public officials. Compared to public officials, male and female HHCWs showed significantly higher SIRs for all cancers. The digestive organ was the most common cancer site among male HHCWs, whereas the breast and female genital organs were the most common cancer sites among female HHCWs. These cancers showed significantly higher SIRs in HHCWs than in public officials.

Epidemiological studies have reported a high cancer risk among all HCWs compared to other medical staff in hospitals. The results of our study showed an increased SIR for all cancers among female HHCWs; however, this has not been consistently reported in previous studies. Rix et al. investigated cancer incidence in Danish HCWs using census data and cancer register records. They studied a total of 13,955 men and 75,052 women during a 17-year period (1970–1987) and found the highest incidence of brain cancer among male doctors (SIR = 2.00; 95% CI = 1.18–3.16) and of breast cancer among female dentists (SIR = 1.63, 95% CI = 1.08–2.35), female doctors (SIR = 1.53; 95% CI = 1.05–2.16), and female registered nurses (SIR = 1.19; 95% CI = 1.08–1.30) [5]. Ekpanyaskul et al. investigated the incidence of cancer in a 14-year retrospective cohort study of 2331 HCWs in Thailand. The SIR of leukemia in female HCWs (SIR = 11.54; 95% CI = 2.38–33.72) and the SIR of all cancers in male physicians (SIR = 6.02; 95% CI = 1.41–19.93) were significantly higher [6]. A study in a university hospital in France constructed a cohort of 940 physicians with a follow-up duration of 10,963 person-years and reported an increased incidence of hematological malignancy (SIR = 5.45; 95% CI = 2.0–11.9) compared to the general population, although the incidence of total cancer was not different from that in the general population (SIR = 0.97; 95% CI = 0.59–1.5) [13]. In Taiwan, a cohort study with 14,889 physicians (follow-up duration of 146,895 person-years) and 29,778 comparisons (292,267 person-years) reported a decreased all-cancer risk (hazard ratio [HR] = 0.86, 95% CI = 0.76–0.97) but an increased prostate cancer risk among men (HR = 1.72, 95% CI = 1.12–2.65) and an increased breast cancer risk among women (HR = 2.00, 95% CI = 1.11–3.62) [14].

In our study, a higher incidence of breast cancer was observed among female HHCWs compared to the general population. Considering that the majority of HHCWs were nurses, these results are consistent with previous reports on breast cancer among nurses. Kjaer et al. investigated cancer incidence in a cohort of 92,140 female Danish registered nurses from 1980 to 2003 and reported a significantly increased SIR of breast cancer (SIR = 1.1, 95% CI = 1.1–1.2) compared with the general population, although the SIR of all cancers was not increased (SIR = 1.0, 95% CI = 0.98, 1.0) [15]. Although the mechanism by which they cause cancer is not entirely understood, the most significant risk factor for breast cancer appears to be estrogen.

Occupational risk factors for breast cancer may include night-shift work and exposure to anti-neoplastic drugs [16]. The International Agency for Research on Cancer states that night-shift work may be carcinogenic to humans (Group 2A); specifically, it may cause breast cancer, as the expression of key circadian genes and serum melatonin are both altered by the light-dark cycle [17]. A systematic review conducted by Fagundo-Rivera et al. also concluded that night shifts were a risk factor for breast cancer among nurses [18]. Nurses are exposed to anti-neoplastic agents, which are potential carcinogens [19,20]. Biological monitoring data showed that HCWs can be exposed to anti-neoplastic agents without being in direct contact [21]. Ratner et al. studied a cohort of 56,213 Canadian nurses during 1974–2000 and reported that nurses who worked in oncology departments showed a significantly higher risk of breast cancer (HR = 1.83, 95% CI = 1.03–3.23) [22]. Shen et al. constructed a cohort of 277,543 HCWs and 555,086 non-HCWs and reported an increased risk of breast cancer among HCWs compared to non-HCWs (HR = 1.34, 95% CI = 1.28–1.41).

Occupational stress can be a carcinogenic risk factor for HCWs. A meta-analysis performed in 2003 to investigate the association between stressful life events and breast cancer risk reviewed 27 relevant studies and concluded that stressful life events were significantly associated with breast cancer risk; however, publication bias was observed [23]. Additionally, the significant effects of job stress on cancer among nurses have yet to be reported. A study that conducted an 8-year follow-up for 37,562 US female registered nurses showed that job strain, measured by Karasek and Theorell’s job content questionnaire, was not associated with an excess risk of breast cancer [24]. In Denmark, in a cohort study with 6571 nurses, a high job-strain group did not show an increased risk of overall cancer or any cancer subtype compared to a low job-strain group (HR = 0.84; 95% CI = 0.7–1.1).

Ionizing radiation is also a potential carcinogen in HCWs. A recent review found 16 epidemiological studies on ionizing radiation exposure among HCWs and concluded that there could be an elevated risk of cancer owing to high and prior exposure to radiation [25]. Lee et al. studied 93,922 medical radiation workers and 827 thyroid cancer cases and reported significantly higher SIRs in male (1.72, 95% CI = 1.53–1.91) and female (1.18, 95% CI = 1.08–1.28) medical radiation workers [26]. In another study on the mortality of medical radiation workers, the SMRs for all causes of death were significantly lower than expected in both men (SMR = 0.45, 95% CI = 0.42–0.48) and women (SMR = 0.49, 95% CI = 0.41–0.58); however, mortality from all cancers (SMR = 1.60, 95% CI = 1.41–1.82), leukemia, colon cancer, stomach cancer, and diseases of the circulatory system increased significantly among male workers [27]. However, in our study, an excess risk of thyroid cancer or leukemia related to ionizing radiation was not found.

In a Korean study, Shin et al. analyzed the differences in mortality between Korean medical doctors and the general population. The SMR for all causes of death among doctors was low (SMR = 0.47, 95% CI = 0.44–0.50) [28]. This pattern was observed not only in Korea but also in other countries. In a study by Danish medical doctors, the mortality rate among doctors was lower than that among the general population. However, in a study conducted by Dimich-Ward et al., the SMR for all causes of death was significantly lower in the female registered nurse group compared to the general population (SMR = 0.61, 95% CI = 0.58–0.64) [22]. In our study, male workers showed a pattern similar to that in this previous study. In male HHCWs, the SMR was significantly lower when the control group was set as the total workers corresponding to the general population. However, when the control group was set as public officials or female workers, the SMR was higher than that reported in these studies. We observed discrepancies in the results when using different reference groups—total workers in Table 2 and public officials as the reference group in Table 3. We believe that these discrepancies may arise because of the differences in the overall health status of the two reference groups, occupational exposure, and lifestyle factors. In our study, there were other discrepancies in the excess cancer risk of some types between the sexes. In male HHCWs, cancers in the respiratory and intrathoracic organs, trachea, malignant neoplasm of the bronchus and lung, and non-Hodgkin’s lymphoma increased. These cancer types did not show a statistically significant difference in SIR in females. Excess SIRs in the breast, mesothelial and soft tissue, and female genital organs were observed only in female HHCWs. This is consistent with the findings of a previous study that the highest causes of cancer-related mortality were bronchial and lung cancer in male HHCWs and breast cancer in female HHCWs [8]. One possible explanation for this phenomenon is the different job distributions between the sexes. In the Western Pacific Region, 60% of physicians are male, and 81% of nurses are female [29]. Different types of cancer with excess risk by gender and disproportional job distribution by gender imply that occupational carcinogens can also differ by the job. Night-shift work is a potential carcinogen associated with cancers of the breast, prostate, colon, and rectum [17]. According to a recent report from the UK, 63% of hospital nurses perform night-shift work [30]. The percentage of shiftwork workers was higher in female HHCWs (mainly nurses) than in male HHCWs. Our results suggest that further investigation is needed to understand the biological and occupational factors causing gender differences in cancer incidence rates among healthcare workers and the impact of job roles and healthcare settings on these disparities.

Because the fixed-job classification of the industry was more conservative, HHCWs could be identified more reliably. There was a significant difference in follow-up duration (person-years) between the fixed and dynamic cohorts. The majority of HHCWs are female; the World Health Organization (WHO) reported that women account for 67% of the global healthcare workforce [29]. However, male workers outnumbered female workers in the dynamic job classification. This difference can be observed because non-regular workers in hospitals are usually male. The dynamic cohort included those who worked for a short period, and the large number of study participants in the dynamic cohort indicates that many people worked for a short period in hospitals. These include waste disposal, facility maintenance, security services, computer system management, expenses, and parking management in hospitals.

Our study is a novel and large-scale study that investigated increased cancer risk among HHCWs in South Korea using representative data for HHCWs in South Korea from the NHIS. However, our study has some limitations. First, there could be a misclassification of cancer cases. A study comparing the Korean Central Cancer Registry and the NHIS showed that the primary diagnosis has a sensitivity of 91.5–97.9% and a positive prediction of 81.8–94.1% [31]. However, we expected that misclassification could lead to a bias in the null hypothesis. Second, information on the specific job in the hospital was not presented. In South Korea, healthcare HHCWs are mainly composed of nurses and doctors [4]; however, future studies are needed to determine which workers in hospitals are more vulnerable to cancer risk. For this purpose, occupational health databases for HCWs are required, such as those in the U.S. or Taiwan [32,33]. Finally, this study only included HHCWs in South Korea. Therefore, further studies are needed to generalize the findings.

## 5. Conclusions

We constructed a cohort of 107,646 HHCWs with a follow-up duration of 905,503 person-years and found an increased incidence of all cancers, including breast cancer, and increased mortality. Increased SIRs for all cancers and breast cancer were noted among female HHCWs. Our data showed that workers in hospitals face a high risk of cancer, which implies that occupational risk factors, such as night-shift work, exposure to antineoplastic drugs and ionizing radiation, and job stress, should be investigated among HHCWs in South Korea. The findings of this study suggest that more detailed research and intervention programs for occupational environments are required.

## Figures and Tables

**Table 1 cancers-15-02045-t001:** Descriptive statistics of the cohort.

	HHCWs	Total Workers	Public Officials
	*n* (%) (or Mean ± SD)	*n* (%) (or Mean ± SD)	*n* (%) (or Mean ± SD)
Total	107,646 (100)	8,317,710 (100)	611,833 (100)
Gender			
Male	37,243 (34.6)	5,740,042 (69.0)	300,080 (49.0)
Female	70,403 (65.4)	2,577,668 (31.0)	311,753 (51.0)
Age at the end of follow-up (yrs)			
25–34	52,469 (48.7)	2,866,046 (34.5)	167,944 (27.4)
35–44	30,738 (28.6)	2,824,660 (34.0)	208,193 (34.0)
45–54	17,316 (16.1)	1,993,575 (24.0)	187,779 (30.7)
55–64	7123 (6.6)	633,429 (7.6)	47,917 (7.8)
Income			
Q1 (lowest)	21,064 (19.6)	1,871,217 (22.5)	17,789 (2.9)
Q2	37,648 (35.0)	1,924,848 (23.1)	45,171 (7.4)
Q3	27,830 (25.9)	2,204,875 (26.5)	194,982 (31.9)
Q4 (highest)	21,104 (19.6)	2,316,770 (27.9)	353,891 (57.8)

HHCWs, healthcare workers in hospitals; SD, standard deviation; Q1–4, Quartile 1–4.

**Table 2 cancers-15-02045-t002:** The observed number of deaths, standardized incidence ratios, and 95% intervals, with total workers as the reference group.

Classification	Male					Female				
	Obs	Exp	SIR	95% CI	*p*	Obs	Exp	SIR	95% CI	*p*
All-cause death	183	219.7	0.83	(0.73–0.96)	0.006	**191**	**153.4**	**1.25**	**(1.06–1.47)**	**0.0019**
[C00–C97] Malignant neoplasm	462	493.2	0.94	(0.86–1.03)	0.082	**1480**	**1360.9**	**1.09**	**(1.03–1.15)**	**<0.001**
[C00–C14] Lip, oral cavity, and pharynx	13	10.0	1.30	(0.71–2.70)	0.208	17	8.3	2.04	(1.05–4.63)	0.005
[C15–C26] Digestive organs	247	250.0	0.99	(0.87–1.12)	0.441	217	194.1	1.12	(0.97–1.29)	0.056
(C16) Stomach	89	95.1	0.94	(0.77–1.16)	0.287	86	80.8	1.06	(0.86–1.34)	0.296
(C18) Colon	45	37.2	1.21	(0.88–1.71)	0.118	43	41.4	1.04	(0.77–1.45)	0.422
(C22) Liver and intrahepatic bile ducts	43	55.1	0.78	(0.60–1.04)	0.055	22	17.4	1.26	(0.79–2.15)	0.162
(C25) Pancreas	8	11.8	0.68	(0.39–1.32)	0.169	16	10.6	1.51	(0.84–3.08)	0.073
[C30–C39] Respiratory and intrathoracic organs	56	48.8	1.15	(0.87–1.55)	0.168	51	41.2	1.24	(0.91–1.72)	0.077
(C33–C34) Trachea and malignant neoplasm of bronchus and lung	48	41.9	1.15	(0.85–1.59)	0.192	46	36.9	1.25	(0.90–1.77)	0.082
[C40–C41] Bone and articular cartilage	3	2.3	1.33	(0.39–9.15)	0.404	4	3.6	1.12	(0.42–4.57)	0.485
[C43–C44] Skin	2	6.4	0.31	(0.15–0.83)	0.046	15	8.7	1.72	(0.90–3.80)	0.033
[C45–C49] Mesothelial and soft tissue	3	5.2	0.58	(0.25–1.75)	0.238	20	9.9	2.02	(1.09–4.22)	0.003
[C50–C50] Breast	0					**376**	**310**	**1.21**	**(1.09–1.36)**	**<0.001**
[C51–C58] Female genital organs						135	116.7	1.16	(0.96–1.40)	0.052
[C60–C63] Male genital organs	21	28.4	0.74	(0.51–1.11)	0.093					
[C64–C68] Urinary tract	36	37.9	0.95	(0.69–1.34)	0.420	21	17.7	1.18	(0.75–2.01)	0.246
[C70–C72] Brain and other parts of the central nervous system	7	7.1	0.99	(0.48–2.46)	0.565	9	12.1	0.75	(0.43–1.44)	0.234
[C81–C96] Lymphoid, hematopoietic and related tissue	39	29.0	1.34	(0.94–2.01)	0.044	47	37.3	1.26	(0.92–1.79)	0.070
(C82–C85) Non-Hodgkin’s lymphoma	19	13.3	1.43	(0.84–2.67)	0.082	21	16.0	1.31	(0.81–2.29)	0.0132
(C91–C95) Leukemia	9	9.3	0.97	(0.52–2.10)	0.548	17	13.3	1.28	(0.75–2.37)	0.187

Obs, observed cases; Exp, expected cases; SIR, standardized incidence ratio; CI, confidence interval. Those with *p* < 0.002 (significant under the Bonferroni correction) are shown in bold.

**Table 3 cancers-15-02045-t003:** The observed number of deaths, standardized incidence ratios, and 95% confidence intervals, with public officials as the reference group.

Classification	Male					Female				
	Obs	Exp	SIR	95% CI	*p*	Obs	Exp	SIR	95% CI	*p*
All-cause death	**183**	**122.4**	**1.50**	**(1.25–1.80)**	**<0.001**	**191**	**117.6**	**1.62**	**(1.36–1.96)**	**<0.001**
[C00–C97] Malignant neoplasm	**462**	**400.4**	**1.15**	**(1.05–1.28)**	**<0.001**	**1480**	**1286.3**	**1.15**	**(1.09–1.22)**	**<0.001**
[C00–C14] Lip, oral cavity, and pharynx	13	8.1	1.61	(0.82–3.70)	0.069	17	8.4	2.02	(1.04–4.55)	0.006
[C15–C26] Digestive organs	**247**	**189.5**	**1.30**	**(1.13–1.51)**	**<0.001**	**217**	**161.7**	**1.34**	**(1.15–1.58)**	**<0.001**
(C16) Stomach	89	74.9	1.19	(0.95–1.51)	0.061	86	69.1	1.24	(0.98–1.60)	0.027
(C18) Colon	45	29.1	1.54	(1.08–2.30)	0.004	43	36.4	1.18	(0.85–1.68)	0.156
(C22) Liver and intrahepatic bile ducts	43	39.9	1.08	(0.79–1.51)	0.332	22	12.4	1.77	(1.02–3.39)	0.009
(C25) Pancreas	8	8.1	0.99	(0.50–2.29)	0.561	16	11.6	1.38	(0.78–2.69)	0.128
[C30–C39] Respiratory and intrathoracic organs	**56**	**30.4**	**1.84**	**(1.29–2.72)**	**<0.001**	51	39.9	1.28	(0.94–1.79)	0.051
(C33–C34) Trachea and malignant neoplasm of bronchus and lung	**48**	**26.1**	**1.84**	**(1.25–2.81)**	**<0.001**	46	35.4	1.30	(0.94–1.86)	0.049
[C40–C41] Bone and articular cartilage	3	1.9	1.59	(0.43–14.45)	0.296	4	2.8	1.45	(0.48–7.77)	0.308
[C43–C44] Skin	2	5.6	0.36	(0.16–1.02)	0.082	15	8.3	1.80	(0.92–4.07)	0.023
[C45–C49] Mesothelial and soft tissue	3	5.1	0.58	(0.25–1.76)	0.251	**20**	**9.1**	**2.19**	**(1.16–4.75)**	**0.001**
[C50–C50] Breast	0					**376**	**319.0**	**1.18**	**(1.06–1.32)**	**0.001**
[C51–C58] Female genital organs						**135**	**90.8**	**1.49**	**(1.21–1.85)**	**<0.001**
[C60–C63] Male genital organs	21	28.3	0.74	(0.51–1.11)	0.096					
[C64–C68] Urinary tract	36	30.3	1.19	(0.83–1.76)	0.171	21	16.1	1.30	(0.80–2.27)	0.137
[C70–C72] Brain and other parts of the central nervous system	7	6.1	1.16	(0.53–3.13)	0.410	9	12.2	0.74	(0.42–1.41)	0.225
[C81–C96] Lymphoid, hematopoietic and related tissue	39	24.1	1.62	(1.09–2.53)	0.003	47	31.0	1.52	(1.07–2.24)	0.004
(C82–C85) Non-Hodgkin’s lymphoma	**19**	**8.7**	**2.19**	**(1.14–4.89)**	**0.0017**	21	11.6	1.80	(1.02–3.53)	0.008
(C91–C95) Leukemia	9	9.9	0.91	(0.49–1.91)	0.471	17	10.1	1.69	(0.92–3.52)	0.029

Obs, observed cases; Exp, expected cases; SIR, standardized incidence ratio; CI, confidence interval. Those with *p* < 0.002 (significant under the Bonferroni correction) are shown in bold.

## Data Availability

The data are not publicly available due to the data belonging to the Korean government.

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
