# Peer review of "Cancer Incidence in Korean Healthcare Workers in Hospitals"

_cancers, 2023, doi:10.3390/cancers15072045_

Round 1

Reviewer 1 Report

Manuscript can be improved by:

Sorting the data and presenting the significant changes in a sperate table. Also highlighting the differences among male and female subjects. 

Please clarify average work duration of the health care workers in relation to observed changes. 

Were there any differences based on the age-groups?

Discrepancies observed should be further discussed for possible reasons. 

Author Response

We appreciate your valuable comments. Please see the attachment.

Reviewer 2 Report

Dear autors,

- please complete the conclusion and results chapter,

- to supplement of references,

- check the formal conditions for publication in the journal,

Author Response

(The authors gave the same response as above.)

Reviewer 3 Report

This is a retrospective cohort of healthcare workers in hospital, during 2005-2017, carried out in South Korea aimed at evaluating overall mortality and 'incidence' of several cancers.

The main point of weakness is that the Authors perform a number of comparisons (vs. all and public office workers) exposing their results to the inflation of alpha error. Therefore, some of the results, including the underlined excesses, may be due to chance. A Bonferroni-like correction should be considered and the point has to be clearly declared and discussed.

The analysed cohort included different professional (and possible exposures). The study would be more interesting splitting the cohort into: doctors, nurses, radiologists, etc.

It is not clear if the HHCW are also included in the group of comparison (all workers). It should not be. If yes, please specified the reasons for this choice and the possible effects on the results. Moreover, why public office workers have been chosen as another reference groups? Please clarify.

Row 82. Are there missing values in the variable identifying the industry classification? Please specify

Row 87. Is it possible to add a reference?

Row 94.  3 years means three consecutive years or 3 during 2005-2017? Please specify

Row 100 The authors speak of cancer incidence. How they detected newly detected cases from prevalent cancers diagnosed before 2005?

Row 119. Please add the legend for Q1 Q4 as in table 1

The sex among the HHCW is strongly unbalanced in favour of women. This is not the case for the dynamic cohort. Could you please explain the reasons for such difference?

Results indicating a decrease in the risk (once corrected for inflation of alpha error) should be commented, including those in the sensitivity analysis.

In table 2 please correct typo: Satndardized Moratlity Ratio 

Author Response

(The authors gave the same response as above.)

Round 2

Reviewer 3 Report

The Authors replied satisfactory to all my questions